# Family Supportive Supervisor Behaviors Moderate Associations between Work Stress and Exhaustion: Testing the Job Demands–Resources Model in Academic Staff at an Austrian Medical University

**DOI:** 10.3390/ijerph19095769

**Published:** 2022-05-09

**Authors:** Nikola Komlenac, Lisa Stockinger, Margarethe Hochleitner

**Affiliations:** Institute of Diversity in Medicine, Medical University of Innsbruck, 6020 Innsbruck, Austria; lisa.naderhirn@gmail.com (L.S.); margarethe.hochleitner@i-med.ac.at (M.H.)

**Keywords:** academic career, publication activity, job demands–resources model, superior family support, work–family services

## Abstract

The time-intensive work of publishing in scientific journals is an important indicator of job performance that is given much weight during promotion procedures for academic positions. The current study applied the job demands–resources model and analyzed whether family supportive supervisor behaviors (FSSB) moderated associations between work stress and feelings of exhaustion as a job resource and whether feelings of exhaustion ultimately mediated the link between work stress and academic employees’ publication activity. The current online cross-sectional questionnaire study was conducted in 133 academic employees (65.4% women, 34.6% men; *M*_age_ = 41.9, *SD* = 10.1) at an Austrian medical university and assessed employees’ numbers of publications, H-index, work stress, feelings of exhaustion, FSSB, and work–family services used. Manifest path models revealed that FSSB moderated the link between experiencing high levels of work stress and strong feelings of exhaustion, especially in employees who had at least one child below the age of 18. Part-time employment was most strongly linked with lower numbers of publications and lower H-index levels. The finding that FSSB acted as a job resource mostly for employees with at least one child below 18 underlines the fact that FSSB is different from other forms of supervisor support. The current study supports recommendations to increase the amount of work–family services and to change organizational norms to be supportive of the successful management of family and work obligations.

## 1. Introduction

Academic university personnel are faced with many obligations, of which teaching, administrative work, and research activities are the main tasks [1]. Medical academic staff may, in addition to those tasks, need to allocate much time to patient care [2]. These high job demands and workloads can cause work stress, as well as conflicts with obligations that extend beyond work (e.g., family obligations), for employees on the academic career track. The experience of frequent work stress and conflicts that arise when work obligations interfere with family responsibilities has in turn been linked to negative health consequences, such as emotional exhaustion [3]. The job demands–resources (JD–R) model [4,5] can explain such associations. According to the model, negative health consequences are likely to develop when job demands are high and when, at the same time, job resources are either limited or not helpful in achieving one’s goals. These associations have been confirmed by empirical studies conducted among academic university personnel [6,7]. Furthermore, the JD–R model predicts a reduced level of job performance as a consequence of poor health [4]. Past studies have reported that academic personnel who experience exhaustion are likely to perceive low levels of their own productivity at work, be dissatisfied with their job, or have intentions to leave the academic career track [3,8,9,10].

The current study adds to the existing body of literature by testing the JD–R model among academic employees at an Austrian medical university and by including an indicator of productivity on which much emphasis is placed when considering academic employees for job progression and promotion, namely, publication activity [11,12,13]. The current study is further unique in that it specifically considers family supportive supervisor behaviors (FSSB; supervisors’ help with managing employees’ conflicting work and family demands) as a potential job resource [14,15].

### 1.1. Job Demands–Resources Model

The job demands–resources (JD–R) model explains the relationships between job demands, resources, experiences of strain and job performance [4,5,16]. According to the model, job demands are physical, social, or organizational aspects of a job that require sustained physical or mental effort and may be experienced as stressful [5]. Job demands can lead to negative health consequences. The feeling of being exhausted is one negative psychological consequence that can negatively impact one’s job performance [4,5,16]. Job resources, on the other hand, are aspects of the job that can be functional in achieving one’s goals or reducing the impact of one’s job demands [5]. Thus, job resources are believed to moderate the association between job demands and experienced strain [4].

The JD–R model has helped explain previously found associations between high job demands and feelings of exhaustion among academic employees at universities. A job demand that is often reported is the large number of tasks that need to be performed in a specified amount of time [3,6]. Additionally, the conflicts that arise from contradictory demands of work and family responsibilities can be stressful to academic employees and thus lead to emotional exhaustion [3,9,17]. Furthermore, not having a secure employment status [7,8], being pressured to increase one’s performance in regard to teaching and publication activity, and being pushed to successfully obtain funding are job demands experienced by academic university personnel [18]. Women are more often affected by the resulting feelings of exhaustion than men [19].

In line with the JD–R model, previous research has supported the link between job resources and reduced feelings of emotional exhaustion. Specifically, perceived support by one’s colleagues or supervisors, supervisory coaching, and perceived institutional support have been shown to be associated with reduced feelings of exhaustion [6,9,18]. These job resources may also help avoid the negative consequences of job demands with regard to academic university personnel’s job performance [4], of which job dissatisfaction, intention to leave the academic career track, or reduced levels of self-perceived productivity have been considered in previous research [3,9,10]. The current study considers a further indicator of academic employee productivity, namely, publication activity.

### 1.2. Academic Careers and Scientific Publications

An academic career starts with an academic position during one’s graduate study. During this time, employees are often either involved in research projects or conduct their own studies. Their goal is to publish research results in scientific journals so that they may obtain the degree of Doctor of Philosophy (Ph.D.) [20,21]. At the Medical University of Innsbruck, one to three publications are often obligatory for obtaining a Ph.D. degree [21]. After receiving their doctoral degree, their academic career entails a period at the assistant or associate level of professorship at a university. To advance further in an academic career, employees need to continue to publish in scientific journals [22]. Only after accumulating a certain number of scientific publications can an employee be considered for a full professorship [23,24]. At the Medical University of Innsbruck, a publication record of at least 15 publications is required to be considered for a professorship [25].

Alongside the number of publications, another indicator is also used to characterize a person’s publication activity, namely, the Hirsch index (H-index) [26]. This index not only includes the number of a person’s publications but also the number of citations. Thus, the H-index gives an idea of the scientific impact of a person’s publications [26].

An employee’s number of publications [11,27,28] and their H-index [29,30,31] are strongly associated with their academic rank [32]. However, compared to men, women have been shown to publish fewer papers in scientific journals [33,34,35,36,37]. Additionally, lower citation rates are reported for women’s publications than for those of men [38,39,40,41].

This gender difference in publication activity contributes to the gender differences that are reflected in regard to academic rank [31]. Women are less likely than men to hold either the position of full professor [29,38,41,42] or a leadership position [43,44]. This gender difference is also evident at the Austrian medical university where this study was conducted [45,46].

### 1.3. Starting a Family and Family-Friendly Policies

For some employees on the academic career track, the time at which the greatest publication activity is required for job advancement coincides with the time when a family is started and with childbearing [23]. Women and men are impacted when starting a family coincides with the need to produce a certain number of publications. However, women are affected more than men [47] because gender norms dictate that the majority of household labor and childcare responsibilities fall to women [48,49]. This unequal allocation of household labor and childcare responsibilities was especially evident during the coronavirus disease 2019 (COVID-19) pandemic [50,51]. Compared to men who had children and worked in academia, women who had children and worked in academia reported that they spent more time on household labor and childcare obligations (including “homeschooling”) during the COVID-19 pandemic than they did before the outbreak [52,53,54].

The need to tend to household obligations can affect the availability of the time that is required for scientific work [55,56]. Thus, time is often seen as a limited but critical resource for the purpose of working on new publications [57]. The successful allocation of enough time to meet the obligations of both family and scientific work can be a challenge for women and men with children [58]. However, women, compared to men, are affected more by the time demands set by household labor and childcare responsibilities, and women are additionally more often willing to scale back to part-time employment or to take a career break than are men [59,60,61,62]. In contrast, to avoid compromising their careers, men are less likely to allocate time for family obligations [59]. Consequently, women who hold academic positions and have children have, on average, less time for their research than do men who hold academic positions and have children [60,63].

The need to balance family and work obligations is sometimes experienced as an impossible task that needs to be individually managed by the employee [64]. In accordance with the JD–R model, the management of conflicting demands from family and work can be experienced as stress and thus lead to a feeling of emotional exhaustion [55,64,65]. The experience of such exhaustion can negatively impact one’s work performance and is associated with reduced levels of publication activity among academic employees [66].

Job resources that might alleviate the links between job demands and feelings of exhaustion include family-friendly policies at the university level [67]. Such policies entail work–family services that focus mostly on the limited time resources of employees with childcare obligations. Thus, most work–family services offer assistance in regard to finding and financing childcare facilities [67,68] (Appendix A).

In addition to the availability of work–family services, immediate supervisors can play a key role in helping employees balance their family and work obligations [69]. A family-supportive supervisor offers emotional, practical, and social support for balancing one’s family and work obligations and is considerate of their employees’ family responsibilities [70,71,72]. A family-supportive supervisor is associated with both a greater willingness to take advantage of work–family services and greater levels of employee satisfaction with regard to balancing work and family obligations [70,73,74,75].

Thus, in accordance with the JD–R model, previous research reports that family supportive supervisor behaviors (FSSB) moderate the association that the experience of conflicting demands of family and work or work-related stressors is seen to have with the experience of exhaustion [76,77]. Namely, the relationship between strong family–work conflicts and felt exhaustion is weaker for employees who perceive their supervisor to be supportive when they need to tend to their family obligations compared to employees who do not receive such supervisor support [77].

In the current study, institutional and supervisor family support is considered instead of other sources of social support, such as support from family members, because increased institutional and supervisor family support has been shown to be more strongly associated with reduced levels of work–family conflict than other sources of social support [78].

### 1.4. Aim of the Current Study

The current study extends the previous research by applying the JD–R model to analyze the links between job stressors, FSSB, feelings of exhaustion, and academic employees’ research activity. The current study is unique in that it analyzed publication activity as an indicator of job performance [4]. This was achieved by considering two indicators of publication activity, namely, one’s number of publications and one’s H-index [26]. The current study is further unique in that it specifically considered FSSB, which is focused on helping employees manage their conflicting work and family demands [14,15]. It was expected that this kind of supervisor support would especially help employees who have childcare obligations.

Thus, based on the JD–R model, the following hypotheses were tested (Figure 1):

**Hypothesis** **1** **(H1).**
*Work stress is associated with feelings of exhaustion. Exhaustion, in turn, is linked to academic employees’ publication activity.*


**Hypothesis** **2** **(H2).**
*FSSB acts as a job resource and*
*moderates the association between work stress and feelings of exhaustion. Furthermore, the use of the medical university’s work–family services is associated with employees’ feelings of exhaustion and their publication activity.*


**Hypothesis** **3** **(H3).**
*FSSB moderates the association between work stress and feelings of exhaustion, especially in employees with children younger than 18, compared to employees with either no children or no children of this age.*


## 2. Materials and Methods

### 2.1. Procedures and Participants

In April 2020, all employees at an Austrian medical university were invited by e-mail to participate in the present online questionnaire study, which was hosted on SoSci: der onlineFragebogen (http://soscisurvey.de/, accessed on 3 April 2020). In June 2020, one additional reminder to participate was sent out to employees. Data collection closed in August 2020. Participants gave informed consent and confirmed that their participation was voluntary. Participation was anonymous, and no participant received any incentive for their participation. The medical university’s ethics committee confirmed that under Austrian law, the current study did not require formal approval by an ethics committee [79,80].

In total, 440 persons took part in the current study. Of the participants, 216 persons were excluded because they stated that they were not employed at the medical university as scientific personnel. Additionally, participants were excluded if they did not respond to almost all questions on the Family Supportive Supervisor Behavior Short-Form (*n* = 51) and the Copenhagen Burnout Inventory (*n* = 18). One participant did not report their age, one participant did not report whether they availed themselves of the university’s work–life balance services, and one participant did not report their gender. Thus, these three participants were also excluded from the analysis. Seventeen responses from employees who were undertaking an internship (and thus were unlikely to be pursuing an academic career) were also excluded. Finally, two participants were excluded from the analysis because they did not report their publication activity. Ultimately, there were *N* = 133 full responses available for the analysis. The results with regard to the H-index were based on 122 responses because 11 persons failed to give information about their H-index.

Due to the abovementioned procedure, it was not possible to obtain exact response rates. However, a response rate of 14.3% could be estimated based on the employment data of the medical university, according to which approximately 926 persons held an academic position in 2020 [81].

The bootstrap method used is known to be reliable and robust in small samples [82]. Thus, the current study’s sample size of 133 was large enough to estimate small to medium associations between the independent variable (stress) and the mediator (exhaustion), as well as small to medium associations between the mediator (exhaustion) and the outcome variable (publication activity), with a statistical power of 0.8 and an α of 0.05 [83,84].

Of the 133 participants, 87 (65.4%) were women, and 46 (34.6%) were men. Sociodemographic information about the sample is presented in Table 1. The participants’ ages ranged from 22 to 70 years. On average, men were older than women (Table 2). Most of the sample held Austrian nationality, were in a relationship at the time of the study, identified as heterosexual, and were employed full-time at the medical university (Table 1). Men more often held a full-time position than women. Nearly half of the sample had at least one child below the age of 18 years, and more than half of the sample held the position of a university assistant (Table 1).

### 2.2. Measures

#### 2.2.1. Sociodemographic Information

The participants were asked for their self-reported gender (response options: woman, man, trans-woman, trans-man, gender-neutral/nonbinary, diverse or other gender identity), age (free-text response), sexual orientation (response options: self-identifying as heterosexual, gay-identified/lesbian-identified, bisexual, asexual, other), relationship status (response options: *single*, *not sure*/*complicated*, *in relationship*), and nationality (response options: *Austrian*, *German*, *Turkish*, *Italian*, *other*). Furthermore, they were asked whether they held an academic position at the medical university (response options: *administrative position*, *academic position*, *grant-financed position*, *do not work at this medical university*), which academic position they held (response options: *internship*, *student assistant*, *lector*, *university assistant*, *tenure track*, *university professor*), and whether they worked full-time (response options: *25%*, *50%*, *70%*, *100%*, *other*). Finally, the participants were asked how many children they had (free-text response). If they reported having children, participants were then asked for their children’s age (free-text response).

If they selected the response of “other” to any question, participants were prompted to provide a free-text response specifying their initial response. In most cases, the free-text responses could be matched to an existing category of a variable. If they could not be matched, free-text responses were excluded. For the analysis, only participants who held an academic position or a grant-financed position were considered. A new variable with two categories was formed for academic position. In this variable, one response category was formed for *student*
*assistants*, *lectors* and *university*
*assistants*, while the second was for *tenure*
*tracks* and *university*
*professors*. A dichotomous variable was formed for employment level that differentiated whether a participant worked full-time (100% or above) or had no full-time employment (all remaining response options). Finally, a new variable was formed with one category (1) for participants who indicated having at least one child younger than 18 years and another category (0) for the remaining participants who had either no children or no children younger than 18 years.

#### 2.2.2. Publication Activity

To assess how many articles participants had published in scientific journals, they were asked the following question: “How many peer-reviewed scientific articles have you authored or coauthored?” [27]. Free-text responses to this question were allotted to four categories, namely, 0 = no publications, 1 = one to three, 2 = four to 15, or 3 = more than 15. This categorization was chosen because one to three publications are often obligatory during graduate study (i.e., to obtain a Ph.D. degree) [20,21]. Another “milestone” in an academic career is to be granted venia docendi, which requires a publication record of at least 15 publications at the medical university [25].

Participants entered their H-index in an open-text field after being asked, “What is your current H-index (https://www.scopus.com/freelookup/form/author.uri, accessed on 29 April 2022)?” The median of the given responses was 8.0 (interquartile range = 0.0–19.2). For the analysis, the responses were grouped into four categories based on quartiles, namely, 0 = no H-index, 1 = one to eight, 2 = nine to 19, and 3 = larger than 19.

#### 2.2.3. Stress at Work

The experience of work stress was measured with a questionnaire developed by Cavanaugh et al. [85]. This questionnaire consisted of 16 statements that each describe challenge stressors (6 items), hindrance stressors (5 items), or other stressors (5 items that do not clearly fall into either category). Challenge stressors are work-related demands that can potentially result in work-related gains for an individual (e.g., “The number of projects and/or assignments I have”). In contrast, hindrance stressors are believed to interfere with being successful at work and are not associated with personal gains (e.g., “The inability to clearly understand what is expected of me on the job”). Using a five-point Likert scale, the participants in the current study reported how much stress each described stressor caused them in their work (1 = produces no stress; 5 = produces a great deal of stress). High mean scores indicated that these stressors caused the participants to experience a great amount of stress at work. A confirmatory factor analysis supported the two-factor structure of the questionnaire [85]. The scale assessing work stress associated with challenge stressors had a reported internal consistency of Cronbach’s α = 0.87, while the scale measuring stress associated with hindrance stressors had a reported internal consistency of Cronbach’s α = 0.75 [85].

After translating all items into German with the back-and-forth procedure [86], an exploratory factor analysis (EFA) with the Kaiser criterion was performed to explore the translated questionnaire’s factor structure in the current sample. The resulting four factors explained 20.5%, 11.9%, 10.6% and 10.6% of the variance. However, items with small communalities (*h*^2^ < 0.40) were detected. The item with the smallest communality (*h*^2^ = 0.18; Item 10) was removed, and the EFA was repeated before the next item with a communality smaller than 0.40 was removed. This process was repeated until five items (Items 7, 8, 10, 12, and 16) were removed and the EFA solution did not include items with communalities smaller than 0.40. The retained items loaded on two factors. The first factor (Items 1, 2, 3, 4, 5, 6, 9, and 13; λ = 0.62–0.81) explained 37.3% of the variance and had satisfactory internal consistency (Table 2) [87]. The second factor (Items 11, 14, and 15) explained 16.8% of the variance. This factor was not considered in the analysis because there was an unsatisfactory internal consistency with regard to men (Table 2) [87].

#### 2.2.4. Feelings of Exhaustion

The Copenhagen Burnout Inventory (CBI) was used to assess participants’ degree of fatigue and exhaustion [88]. The six items of the first scale, namely, Personal Burnout, ask persons how tired or exhausted they felt, e.g., “How often are you physically exhausted?” Responses were given on a five-point Likert scale (1 = Never/almost never; 5 = Always). The seven items of the Work-related Burnout Scale assessed the degree of fatigue and exhaustion that participants perceived in relation to their work, e.g., “Do you feel burnt out because of your work?” Responses were given on a five-point Likert scale (1 = Never/almost never or to a very low degree; 5 = Always or to a very high degree). High mean scores indicated that a person experienced high degrees of fatigue and exhaustion [88]. No factor analyses were performed in the original study, but satisfactory internal consistencies were reported for the two scales (Cronbach α = 0.87).

All items were translated into German by two independent professional translators with the back-and-forth procedure [86]. An EFA with the Kaiser criterion supported a two-factor structure (factors explained 28.6% and 18.7% of the variance). However, items with small communalities (*h*^2^ < 0.40) were detected. The item with the smallest communality (*h*^2^ = 0.19; Item 11) was removed, and the EFA was repeated before the next item with a communality smaller than 0.40 was deleted. This process was repeated until seven items (Items 6, 8, 9, 11, 12, and 13) were removed and the EFA solution did not include items with communalities smaller than 0.40. All of the retained items loaded on one factor (λ = 0.63–0.86) that explained 51.0% of the variance and resembled the original Personal Burnout Scale [88]. The items measured perceived fatigue and exhaustion with a satisfactory level of internal consistency (Table 2) [87].

**Table 2 ijerph-19-05769-t002:** Descriptive statistics.

Variable ^1^	All			Women			Men			*t*(131)	*d*
	*M*	*SD*	α	*M*	*SD*	α	*M*	*SD*	α		
Age	41.9	10.1		40.0	9.1		45.6	11.1		3.1 *	0.57
Stress 1	2.9	0.9	0.89	2.8	0.9	0.88	3.0	1.0	0.91	1.1	
Stress 2	2.8	1.1	0.75	3.0	1.1	0.80	2.6	0.9	0.59	−1.8	
Exhaustion	2.7	0.7	0.88	2.8	0.7	0.89	2.6	0.7	0.86	−1.5	
FSSB-SF	3.1	1.0	0.89	3.1	1.1	0.89	3.0	1.0	0.88	−0.5	

^1^ All Likert scales ranged from 1 (disagreement/no stress) to 5 (agreement/a lot of stress); FSSB-SF = Family Supportive Supervisor Behavior Short-Form; df = degrees of freedom; * *p* < 0.01.

#### 2.2.5. Family Supportive Supervisor Behaviors

The degree to which superiors supported their employees’ family roles and thus supported employees in their balancing work and family responsibilities was assessed with the Family Supportive Supervisor Behavior Short-Form (FSSB-SF) [14]. The FSSB-SF consists of four items that were retained from the original four-factor structured Family Supportive Supervisor Behaviors scale (FSSB) [15]. The FSSB-SF contains one item of each of the theoretically derived factors, namely, emotional support, instrumental support, role modeling, and creative work–family management [14,89]. One example item is “Your supervisor makes you feel comfortable talking to him/her about your conflicts between work and non-work” (emotional support). Responses were given on a five-point Likert scale (1 = Strongly disagree, 5 = Strongly agree), and mean scores were calculated. High mean scores indicated that employees perceive that their superior(s) support them in balancing their work and family responsibilities. The reported internal consistency was α = 0.82–0.88 [14].

For the current study, the items were translated into German with the back-and-forth procedure [86]. An EFA with the Kaiser criterion revealed a one-factor solution [90]. The proportion of variance explained by this factor was 66.5% (all communalities >0.55), and all items loaded with factor loadings between 0.74 and 0.88. The internal consistency of the FSSB-SF was satisfactory (Table 2) [87].

#### 2.2.6. Use of Work–Family Services

Participants were asked how often (0 = never; 1 = once; 2 = sometimes; 3 = often) they take advantage of each of the work–family services offered by the medical university [67,68] (Appendix A). Because few employees used the response options 2 = sometimes and 3 = often, new variables with two categories were formed for each work–family service. Each new variable contained information on whether an employee had used a certain work–family service (1 = yes, at least once; 0 = no) [72,77]. For the analyses, a dichotomous variable was used to differentiate whether participants had used the university’s work–family services at least once or had not used such services.

### 2.3. Statistical Analysis

To report the descriptive statistics, the percentages and means of the given responses were calculated. Chi-square tests and *t*-tests were used to reveal gender differences. Correlation analyses were performed to calculate the bivariate relationships between the studied variables.

EFAs with the Kaiser criterion (Eigenvalue for factor extraction ≥1) were calculated to analyze whether the factor structures of the validated questionnaires were also applicable to the translated versions of the questionnaires in the current sample. Therefore, items with small communalities (*h*^2^ < 0.40) were removed, and scales consisting of items with high factor loadings and satisfactory internal consistencies were used for the subsequent analyses [87,90].

**Figure 1 ijerph-19-05769-f001:**
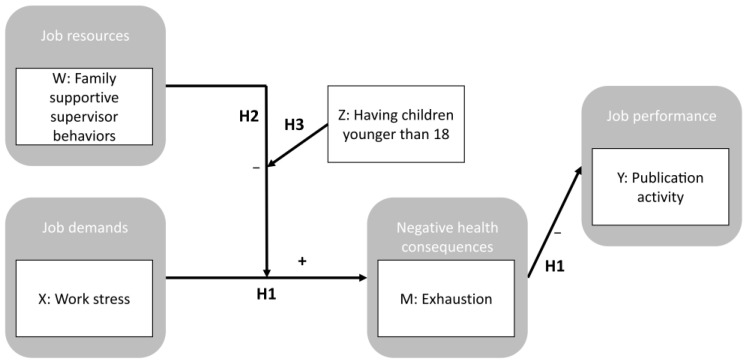
Part of the Job Demands–Resources Model Considered in the Current Study. Based on the Job Demands–Resources Model, it was tested whether work stress (X) would be associated with feelings of exhaustion (M) and, consequently, be linked to academic employees’ publication activity (Y; H1). Family supportive supervisor behaviors (W) were tested as a moderator of the association between work stress and feelings of exhaustion (H2). Finally, it was tested whether family supportive supervisor behaviors would moderate the association between work stress and feelings of exhaustion, especially in employees with children younger than 18 (Z; H3). Arrows represent associations between variables. Higher scores in one variable were expected to go along with higher scores in another variable (“+”) or with a lower score in another variable (“−“).

To test the job demands–resources model, two manifest path models were calculated (Figure 1). In one model, the number of peer-reviewed publications (variable Y_1_) was predicted by feelings of exhaustion (mediator, variable M) and stress at work (variable X). In the second model, the H-index (variable Y_2_) was predicted by those same variables. In both models, feelings of exhaustion were entered as a mediator of the relationship between stress at work and publication activity. FSSB (moderator, variable W) was analyzed as a moderator of the relationship between stress at work and publication activity, as well as stress at work and feelings of exhaustion. Having at least one child below the age of 18 (moderator, variable Z) was entered as a moderator of the relationship between stress at work and publication activity, as well as stress at work and feelings of exhaustion. Finally, indirect links between stress at work and publication activity via feelings of exhaustion were calculated. In all analyses, the control variables of gender, age, employment level, academic position and the use of work–family services were considered. Each of the manifest path models was calculated with Model 12 in the PROCESS macro for SPSS [82] (www.processmacro.org, accessed on 21 January 2021). For this purpose, bootstrap bias-corrected 95% confidence intervals (bootstrap sample was *n* = 5000) for all path coefficients were estimated. Significant results were indicated when *p* ≤ 0.05 or when 95% confidence intervals did not include zero. All analyses were performed with the Statistical Package for the Social Sciences (SPSS) for Windows, version 26.0 (IBM Corp., Armonk, NY, USA).

## 3. Results

### 3.1. Descriptive Statistic

On average, women and men reported experiencing moderate levels of stress at work and moderate feelings of exhaustion (Table 2). Women and men perceived moderate levels of FSSB on average (Table 2). Women with at least one child below the age of 18 more often availed themselves of at least one work–family service (85.4%) than did men with at least one child below the age of 18 years (50.0%), χ^2^(1) = 9.1, *p* = 0.006, V = 0.38. Thereby, the service most frequently used consisted of financial services when resuming work after parental leave (Appendix A). Women more often took advantage of this so-called “Back to Work Campaign” than did men (Appendix A). The second most frequently sought service was help in finding childcare, followed by participation in events and courses for children (Appendix A).

Nearly half of the participants reported having published 16 or more scientific articles (Table 1), although men more often reported having published this number of scientific articles than did women (Table 1). Women more often than men reported not having published any scientific article or having published four to 15 articles (Table 1). This gender difference in publication was also reflected in the H-index results. Women reported having lower H-indices than those of men (Table 1).

Bivariate correlations of the variables are reported in Table 3. Participants’ older age correlated with a larger number of publications and a higher H-index. For women, a higher H-index was additionally associated with having full-time employment. For men, a higher H-index was associated with holding a higher academic position. Having at least one child below 18 years of age correlated with experiencing higher levels of exhaustion and being less frequently employed full-time for women but not for men. For women, a higher position was linked to having at least one child below 18 years of age, using work–family services, and not having full-time employment. Experiencing frequent stress at work was linked to having stronger feelings of exhaustion for women and men and to having higher levels of publication activity for men (Table 3).

### 3.2. Job Demands and Resources

All path coefficients of the manifest path analysis are reported in Table 4. The experience of frequent exhaustion was linked to perceiving little FSSB and to frequent experiences of stress at work (Table 4). Work–family services were especially used by employees who experienced frequent exhaustion. FSSB more strongly moderated the association between stress at work and the experience of exhaustion in employees with at least one child younger than 18 years of age (Figure 2) than it did in employees with either no children or with children older than 18 years of age. In employees with at least one child younger than 18 years of age, the association between experiences of stress and exhaustion was markedly weaker in employees who perceived FSSB than in employees who did not perceive FSSB (Figure 2).

Having a larger number of publications and a higher H-index was linked with being employed full-time and with being older. No other variables were linked to either the publication number or the H-index (Table 4). No indirect associations between stress at work and publication number via exhaustion were detected (all β values = 0.0–0.1; lower limit of 95% confidence interval = −0.1–0.0; upper limit of 95% confidence interval = 0.1–0.2). Similarly, no indirect links between stress at work and the H-index via exhaustion were found (all β values = −0.1–0.0; lower limit of 95% confidence interval = −0.3–−0.1; upper limit of 95% confidence interval = 0.0).

## 4. Discussion

The current study applied the job demands–resources (JD–R) model [4,5,16] and revealed links between high levels of work stress and frequent feelings of exhaustion (H1) among academic employees at an Austrian medical university. Family supportive supervisor behaviors (FSSB) moderated the association between work stress and feelings of exhaustion, especially in employees with children younger than 18 years compared to employees who had either no children or no children of this age (H3). Employees who experienced frequent exhaustion were more likely to use work–family services than were employees who experienced exhaustion infrequently (H2). In contrast to expectations (H1), neither work stress nor feelings of exhaustion were linked to publication activity in the tested models. Instead, part-time employment was most strongly linked with lower numbers of publications and lower H-indices.

### 4.1. Work Stress and Exhaustion

Women in academic positions often view the balancing of time resources between family and work obligations as an additional, personal, and sometimes impossible task that successful employees are expected to fulfill [64]. Thus, the management of work and family obligations can sometimes be seen as an additional work stressor, especially for women who retain the major share of household and childcare obligations [91]. In the current study, having at least one child below the age of 18 was linked to the experience of high levels of exhaustion in women but not in men. The current study revealed that most employees who experienced high levels of exhaustion took advantage of institutional work–family services, such as on-site childcare or affordable childcare options. The important role of such institutional family support in the reduction of work–family conflict [78] and in increasing the retention rate of employees on the academic career track, especially among women, has been previously reported [92,93].

However, previous research has also reported that interventions need to focus not merely on the development and implementation of formal family-friendly policies but also on changing the perception and organizational norms or values surrounding the management of family and work obligations. Thereby, immediate supervisors can play a key role by offering emotional, practical, and social support to employees in regard to balancing their family and work obligations and by being considerate of their employees’ family responsibilities [70,71,72,77]. In line with the JD–R model, the current study shows that such family supportive supervisor behaviors can act as a job resource because FSSB moderates the links between work stress and feelings of exhaustion, especially in academic employees with at least one child below the age of 18. Specifically, in employees with at least one child younger than 18 years of age, the association between high levels of work stress and high levels of exhaustion was weaker when employees perceived high levels of FSSB compared to employees who did not perceive support from their immediate supervisor in regard to balancing their work and family responsibilities.

The finding in the current study that FSSB was a moderator mostly for employees with at least one child under 18 years of age underlines the fact that FSSB differs from other forms of supervisor support [14,89]. Therefore, it is recommended that supervisors either learn or be trained how to specifically support their employees in regard to balancing their work and family responsibilities [69,94].

### 4.2. Publication Activity (Job Performance)

Having a certain number of publications with a certain scientific impact is relevant because publication activity is an important criterion for promotion in academic medicine [32], including at the medical university in this study [21,25]. Such criteria for promotion, however, can place women at a disadvantage because women often have a smaller number of publications and, on average, have publications that are less frequently cited than those of men [31,36,39]. This gender difference was also evident in the current study. More men than women reported having more than 16 publications and having an H-index greater than 20.

Time is a critical resource for being able to work on scientific projects and new publications [57]. The huge block of time required for the publication activity can conflict with the time demands set by household obligations [55]. Gender differences in publication activity can in part be explained by different time demands set by household obligations for women and men. Gender norms often dictate that the majority of household labor and childcare responsibilities fall to women [48,49], and in addition, women are more often willing to scale back to part-time employment compared to men [59,60,62]. This gender difference was also evident in the current study. Women were more often employed part-time, whereas men more often held full-time positions. Consequently, on average, women in academic positions had less time at their disposal for their research compared to that of men [60,63]. Full-time employment in turn was most strongly associated with high levels of publication activity in the current study.

To help with the conflicting time demands set by family and work obligations, family-friendly policies have been enacted at the Medical University of Innsbruck [67]. In the current study, women holding part-time employment have taken special advantage of the work–family services granted under those polices. Thus, the current findings suggest that work–family services can help employees delegate more time to the important and valued task of scientific publishing [57]. Such offers can help employees to better succeed in their job, and women have reported that work–family services concerning childcare and opportunities for work–life integration also motivate them to continue working in their academic position [93]. Therefore, the implementation of work–family services that help find and finance childcare is recommended for universities that do not have such offers [56,95].

### 4.3. Future Directions

Even though time is a critical resource for an academic employee’s publication activity [57], and women can often allocate less time to research than men [60], additional factors have been found to contribute to the gender differences found in regard to publication activity. The stereotypical ideas of gender and the perception of a typical scientist being a man [96] often lead to implicit (often unconscious and unintended) biases that are characterized by women’s scientific contributions being systematically valued less than those of men [97]. Women’s opportunities to conduct planned studies, which form the basis for scientific publications, are hampered by their grant applications being systematically perceived as being of lesser quality than those of men [98,99]. For example, among applications for grants at the Canadian Institutes of Health Research, women who were equally as qualified as men (based on their past research records) received lower application scores than did male applicants [98]. Additionally, women’s scientific texts are often judged as being of poorer quality than equal contributions authored by men [100]. It was found that people judge identical scientific texts differently solely on the basis of the author’s name, i.e., whether the name indicates a female or male author [101]. Receiving less credit for their publications places women at a disadvantage in their job progression and their chances for promotion, even though women can show an equal number of publications as their male contenders when they are up for a potential promotion [31,97,102,103].

Another approach to reducing the disadvantages that women face during their career advancement path may be to not focus on helping women allocate more time to work on scientific publications. Rather, considerations seem warranted to change the standards for promotion to focus less on publication activity and more on alternative achievements [28,104]. For example, in promotion procedures, greater weight could be given to non-research activities in which women are often strongly involved, such as teaching, patient care, or committee work [28,104,105].

### 4.4. Limitations

Several limitations of the current study should be considered. Data collection for the study coincided with the COVID-19 outbreak in Europe [106]. Government regulations led to the temporary closure of many childcare facilities and schools [107]. This led to increased childcare demands for persons with childcare obligations, and women were especially affected [52,53,54]. More women than men experienced the need to tend to children who were either not cared for or did not undergo schooling outside the home as new stressors and as new challenges to balancing their family and work obligations [52,53,54,107]. Consequently, women’s research productivity during this time period was particularly low [108]. The current study was conceptualized before the COVID-19 outbreak and, therefore, did not assess which new challenges persons with childcare obligations faced during lockdown. Some of the exhaustion and work stress reported by the current participants may be attributed to the novel challenges they faced during the COVID-19 outbreak and lockdowns.

The challenging COVID-19 times may also have affected employees’ willingness to participate in the current study. The estimated response rate of 14.3% was relatively low. Even though the sample size seemed adequate for data analysis [82,83], such a relatively small sample size limits the generalizability of the study’s results. Furthermore, the sample was not large enough to calculate separate models for women and men or to consider gender as a moderator (in an interaction term). The relatively small sample also necessitated considering women and men as monolithic groups. However, only the consideration of intragender variations (for example, by additionally considering women’s and men’s nationality, sexual orientation, or relationship status) can help reveal the experiences of overlapping dynamics of different forms of inequality, in the face of which certain identities may be especially advantaged or disadvantaged [63].

Because of the cross-sectional study design with only one time point, no conclusions about the directionality of the found associations or causality can be drawn from the current study’s results, and associations should not be misinterpreted as causal relationships. As is the case with many questionnaire studies, the participants may have given some socially desirable responses and thereby biased the current study’s results. A related source of bias is the so-called common method bias [109]. This bias describes a situation in which associations between (latent) constructs that have been assessed with similar methods (e.g., questionnaires with the same scale format) might in part correlate because of the common method used to assess the studied constructs. However, in the current study, not only associations between latent constructs that share common methods were studied. Instead, some included variables, such as the number of children younger than 18 years of age, the number of publications or one’s H-index, that are objective information, and the assessment of those variables differed from the assessment of latent constructs. For the reporting of their H-index, participants were even provided with an internet link to be able to give accurate responses.

Thus, the assessment of the H-index can be understood as a “fact-based questionnaire item” [110] that can give an estimate of the scientific impact of a person’s publications by considering their number of publications and their number of citations [26]. However, there are several pitfalls related to using the H-index that limit the interpretation of the current study’s findings [111]. First, there can be long delays between the time an author finishes a manuscript and the time at which the manuscript is published. Thereafter, more time passes before the author’s number of citations increases. Thus, it is unlikely that young researchers or researchers at the beginning of their career have a high H-index, and it is possible that a part of our participants’ publication activity was missed. Other factors, such as the popularity of the topic, the number of collaborations, or the number of self-citations, influence a researcher’s H-index. Finally, a researcher might have impacted the scientific field by a few highly cited publications without having an overall high number of publications. In such a case, their H-index would nevertheless remain low [111].

## 5. Conclusions

The current study revealed that family supportive supervisor behaviors (FSSB) can be considered a job resource among employees who hold an academic position at an Austrian medical university. As predicted by the JD–R model [4,5], FSSB acted as a moderator for the link between experiences of work stress and feelings of exhaustion, especially for employees who had at least one child below the age of 18 years. In employees with at least one child younger than 18 years, the association between high levels of work stress and strong feelings of exhaustion was markedly weaker in employees who perceived FSSB than in employees who did not perceive such support. The current study’s findings suggest that work–family services might help employees delegate more time to the important task of scientific publishing and thus support recommendations to increase (or at least maintain) the scope of work–family services offered by family-friendly institutional policies. Additionally, supervisors need to be trained on how to offer emotional, practical, and social support with a view to balancing the family and work obligations of their employees in order for such a family-friendly organizational climate to develop [70].

Future studies need to focus on finding job resources that can help academic employees continue in a full-time position after starting a family. In particular, women have been reported to be willing to drop down to part-time from full-time employment after having a child [60,62]. Future studies should also consider other elements of the JD–R model that were not considered in the current study, such as employee motivation or active changes that employees can make in their job demands and resources (so-called “job crafting”) [4]. In addition to publication activity, other indicators of job performance can be considered in future studies. Receiving research funding, receiving recognition awards, being invited to speak at scientific conferences, or being on the editorial boards of scientific journals are additional indicators of job performance that are given considerable weight in promotion procedures for academic positions [104] and should, therefore, be included in future studies.

## Figures and Tables

**Figure 2 ijerph-19-05769-f002:**
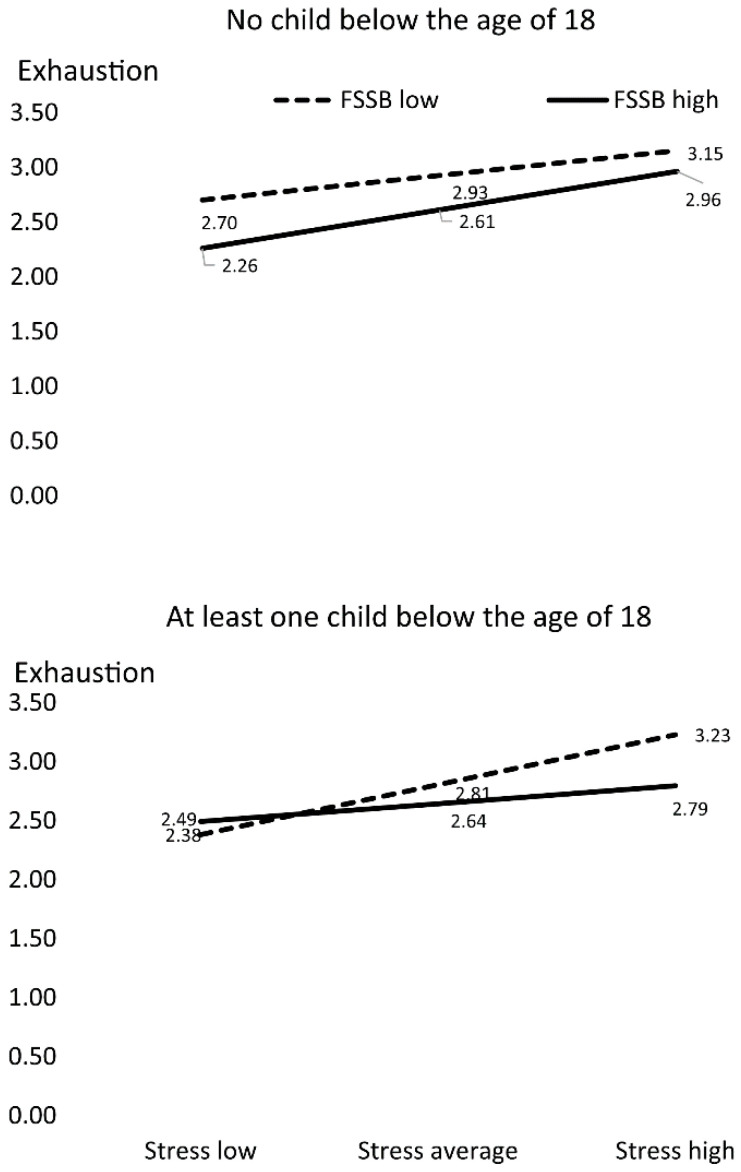
Interaction Stress × Family Supportive Supervisor Behaviors × Child. Dotted lines show employees who, on average, do not perceive family supportive supervisor behaviors (FSSB). Solid lines show employees who, on average, perceive FSSB. Strong work stress was associated with strong feelings of exhaustion. On average, employees who perceived FSSB experienced weaker feelings of exhaustion than did employees who did not perceive FSSB. FSSB more strongly moderated the association between stress at work and the experience of exhaustion in employees with children below the age of 18. In employees with children younger than 18, the association between work stress and exhaustion was weaker for employees who perceived FSSB than for employees who did not perceive FSSB.

**Table 1 ijerph-19-05769-t001:** Sociodemographic description of the sample.

Variable		All *N* (%)	Men *n* (%)	Women *n* (%)	χ^2^	df	*V*
Gender							
	Woman	87 (65.4)					
	Man	46 (34.6)					
Nationality					0.6	3	
	Austrian	91 (68.4)	32 (69.6)	59 (67.8)			
	German	19 (14.3)	6 (13.0)	13 (14.9)			
	Italian	19 (14.3)	6 (13.0)	13 (14.9)			
	Other	4 (3.0)	2 (4.3)	2 (2.3)			
Relationship					0.3	1	
	Single	11 (8.5)	3 (6.5)	8 (9.5)			
	With partner	119 (91.5)	43 (93.5)	76 (90.5)			
Sexual Orientation					2.6	2	
	Heterosexual-identified	125 (94.0)	44 (95.7)	81 (93.1)			
	Gay/lesbian-identified	4 (3.0)	2 (4.3)	2 (4.3)			
	Bisexual-identified	4 (3.0)	0 (0.0)	4 (4.6)			
Child under age of 18					0.0	1	
	No	70 (52.6)	24 (52.2)	46 (52.9)			
	Yes	63 (47.4)	22 (47.8)	41 (47.1)			
Work–family service used					3.5	1	
	No	81 (60.9)	33 (71.7)	48 (55.2)			
	Yes	52 (39.1)	13 (28.3)	39 (44.8)			
Full-time employment					6.1 *	1	0.22
	No	38 (28.6)	7 (15.2)	31 (35.6)			
	Yes	95 (71.4)	39 (84.8)	56 (64.4)			
Academic position					0.1	1	
	University assistant	79 (59.4)	27 (58.7)	52 (59.8)			
	Professor/tenure track	54 (40.6)	19 (41.3)	35 (40.2)			
Publications					9.7 *	3	0.27
	0	18 (13.5)	3 (6.5)	15 (17.2)			
	1–3	17 (12.8)	4 (8.7)	17 (12.8)			
	4–15	24 (18.0)	5 (10.9)	24 (18.0)			
	16+	74 (55.6)	34 (73.9)	74 (55.6)			
H-index					8.2 *	3	0.26
	0	26 (21.3)	6 (14.6)	20 (24.7)			
	1–8	32 (26.2)	8 (19.5)	24 (29.6)			
	9–19	30 (24.6)	9 (22.0)	21 (25.9)			
	20+	34 (27.9)	18 (43.9)	16 (19.8)			

df = degrees of freedom; * *p* < 0.001.

**Table 3 ijerph-19-05769-t003:** All correlation coefficients.

Variables	1	2	3	4	5	6	7	8	9	10
1. Age	-	0.26 *	0.35 **	−0.08	0.18	0.08	0.03	0.13	0.48 **	0.56 **
2. Child under age 18	−0.27	-	0.77 **	−0.40 **	0.21 *	0.02	0.31 **	−0.14	0.13	−0.01
3. Work–family services used	−0.13	0.46 **	-	−0.34 **	0.35 **	−0.04	0.21	−0.02	0.15	0.06
4. Full-time employment	−0.01	−0.08	0.13	-	−0.22 *	0.17	-0.03	−0.04	0.17	0.24 *
5. Academic position	0.11	−0.10	0.06	0.11	-	0.00	0.07	0.14	0.13	0.14
6. Stress	−0.20	0.06	0.03	0.06	0.00	-	0.41 **	0.02	0.13	0.09
7. Exhaustion	−0.30 *	−0.10	0.28	0.22	0.17	0.49 **	-	−0.19	0.08	−0.15
8. FSSB-SF	−0.14	0.05	−0.07	−0.16	−0.06	0.10	−0.13	-	0.05	0.12
9. Publications	0.33 *	−0.12	0.12	0.11	0.10	0.31 *	0.25	0.20	-	0.76 **
10. H-index	0.62 **	−0.13	0.08	0.16	0.35 *	−0.05	−0.08	0.04	0.80 **	-

Above the diagonal, correlations in women (*n* = 100) are reported; below the diagonal, correlations in men (*n* = 50) are reported; FSSB-SF = Family Supportive Supervisor Behavior Short-Form; * *p* < 0.05, ** *p* < 0.01.

**Table 4 ijerph-19-05769-t004:** All model path coefficients—direct associations.

Outcome Variable	Predictor	*B*	*SE B*	95% CI for *B*		*R* ^2^
				LL	UL	
Feelings of exhaustion						0.35 ***
	Gender	0.18	0.12	−0.05	0.41	
	Age	−0.01	0.01	−0.02	0.00	
	At least one child below 18	−0.04	0.14	−0.32	0.23	
	Full-time employment	0.00	0.12	−0.24	0.24	
	Academic position	0.08	0.11	−0.13	0.29	
	**Work–family services used**	0.38 **	0.14	0.10	0.67	
	**Stress**	0.31 ***	0.06	0.20	0.42	
	**FSSB**	−0.12 *	0.05	−0.22	−0.02	
	FSSB × Child	0.08	0.10	−0.12	0.27	
	Stress × FSSB	−0.03	0.05	−0.13	0.07	
	FSSB × Child	0.08	0.10	−0.12	0.27	
	**Stress** × **FSSB** × **Child**	−0.21 *	0.10	−0.41	−0.01	
Publications						0.31 ***
	Gender	−0.22	0.20	−0.61	0.17	
	**Age**	0.05 ***	0.01	0.03	0.06	
	At least one child below 18	0.09	0.23	−0.37	0.56	
	**Full-time employment**	0.48 *	0.20	0.08	0.89	
	Academic position	0.10	0.18	−0.26	0.46	
	Work-family services used	0.11	0.25	−0.39	0.60	
	Exhaustion	0.16	0.15	−0.15	0.47	
	Stress	0.15	0.11	−0.06	0.36	
	FSSB	0.12	0.09	−0.05	0.29	
	Stress × Child	0.07	0.19	−0.31	0.45	
	Stress × FSSB	0.10	0.09	−0.07	0.27	
	FSSB × Child	−0.09	0.17	−0.42	0.24	
	Stress × FSSB × Child	−0.11	0.17	−0.45	0.24	
H-index						0.48 ***
	Gender	−0.04	0.18	−0.40	0.32	
	**Age**	0.06 ***	0.01	0.04	0.08	
	At least one child below 18	0.00	0.22	−0.44	0.43	
	**Full-time employment**	0.64 **	0.19	0.27	1.01	
	Academic position	0.30	0.17	−0.03	0.63	
	Work–family services used	0.04	0.23	−0.42	0.50	
	Exhaustion	−0.22	0.14	−0.50	0.07	
	Stress	0.12	0.10	−0.08	0.32	
	FSSB	0.07	0.08	−0.08	0.23	
	Stress × Child	0.01	0.18	−0.35	0.36	
	Stress × FSSB	0.06	0.08	−0.10	0.22	
	FSSB × Child	−0.28	0.16	−0.59	0.03	
	Stress × FSSB × Child	−0.06	0.16	−0.38	0.26	

Variables that have significant associations with the outcome variable are written in bold. CI = confidence interval; FSSB = family supportive supervisor behaviors; Child = at least one child below 18; * *p* ≤ 0.05, ** *p* ≤ 0.01, *** *p* ≤ 0.001.

## Data Availability

The data presented in this study are available on request from the corresponding author.

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
