# Peer review of "Family Supportive Supervisor Behaviors Moderate Associations between Work Stress and Exhaustion: Testing the Job Demands–Resources Model in Academic Staff at an Austrian Medical University"

_ijerph, 2022, doi:10.3390/ijerph19095769_

Round 1
Reviewer 1 Report
Dear authors,
Thanks for the opportunity to review your paper "Family Supportive Supervisor Behaviors Moderate Associations between Work Stress and Exhaustion: Testing the Job Demands - Resources Model in Academic Staff at an Austrian Medical University." It is a paper that focuses the reader's attention on a theme significant for academics. The study and all formulated hypotheses are well-grounded, and the authors describe the methodology adequately, allowing for the reproduction of the study. Although analyzing figure 1, the results in table 4, and considering the use of the "Model 12 in the PROCESS" (line 384), I wonder if, when the authors state, "FSSB (moderator, variable W) was analyzed as a moderator of the relationship between stress at work and publication activity, as well as feelings of exhaustion and publication activity," in lines 376 and 378, they did not mean to say "FSSB (moderator, variable W) was analyzed as a moderator of the relationship between stress at work and publication activity, as well as stress at work and feelings of exhaustion."
The result, the discussion, and the conclusion are coherent and allow answers to the hypotheses in the research. The authors should highlight the next step of their investigation: how these results will be used and how they are relevant for others to read. I believe that the authors can also improve the abstract to increase the readers' motivation and give credit to the mediation and moderation analysis that developed the novelty and the study's main findings.
Author Response
We would like to thank the Reviewer for their valuable comments as well as for their suggestions and detailed instructions on how to improve our manuscript. We are very grateful for all the time and effort expended in reviewing our manuscript. We have attempted to apply and make changes to the manuscript according to your comments. Please find below our detailed responses.
1) Although analyzing figure 1, the results in table 4, and considering the use of the "Model 12 in the PROCESS" (line 384), I wonder if, when the authors state, "FSSB (moderator, variable W) was analyzed as a moderator of the relationship between stress at work and publication activity, as well as feelings of exhaustion and publication activity," in lines 376 and 378, they did not mean to say "FSSB (moderator, variable W) was analyzed as a moderator of the relationship between stress at work and publication activity, as well as stress at work and feelings of exhaustion."
Response 1: Thank you very much for directing our attention to this error. This sentence has been changed as you suggested.
2) The authors should highlight the next step of their investigation: how these results will be used and how they are relevant for others to read.
Response 2: We agree that future directions and the next steps to be investigated need to be offered. (1) We discuss directions for potential future studies in the Conclusion (Lines 636ff; e.g., to include other components of the job demands - resources model, especially so-called “job crafting”). (2) In the Discussion section we conclude that that FSSB differs from other forms of supervisor support and, thus, that supervisors need to know how to be family supportive (Line 504). (3) We stress the benefits of family-friendly policies, especially work-family services that help find and finance childcare and recommend such services for universities that do not have such offers (Lines 531). (4) In Chapter 4.3. Future Directions (Lines 524ff) we discuss that (implicit) gender bias and gender stereotypes need to be given further consideration.
3) I believe that the authors can also improve the abstract to increase the readers' motivation and give credit to the mediation and moderation analysis that developed the novelty and the study's main findings.
Response 3: Thank you for giving us suggestions on how to attract the attention of a larger number of readers. In the Abstract we now more strongly stress that we conducted a mediation and moderation analysis. We already mentioned in the Results section of the Abstract that we found a significant 3-way interaction (Stress x Family Supportive Supervisor Behaviors x Child). In the Conclusion of the Abstract we stress that this 3-way interaction (i.e., moderation) indicates that FSSB might be different from other forms of supervisor support.
We hope you will find that your comments are reflected in our revised manuscript. We are very grateful for your detailed feedback and instructions on how to improve our manuscript. We have attempted to incorporate all your suggestions. In cases where we were not able to exactly follow your suggestions, we hope you approve of our arguments and approach.
Reviewer 2 Report
Strengths: literature review, methodology, analysis of results
Weaknesses: the sample
Improvement proposals
Line 89 “Only after accumulating a certain number of scientific publications can an employee be considered for a full professorship [23,24].” Authors should specify, if possible, the minimum number of scientific publications.
Line 100 – “Women are less likely than men to hold either the position of full professor 8,37,40,41] or the leadership position [42,43]. Authors must describe the reasons or circumstances for this to happen.
Line 481 – “The current study revealed that most employees who experienced high levels of exhaustion took advantage of institutional work” Authors should specify, if possible, in concrete values.
Line 507- “Having a certain number of publications with a certain scientific impact is relevant publication because activity is an important criterion for promotion in academic medicine. Authors should specify, in concrete numbers, the number of publications for academic relevance.
Line 546 - “Additionally, women’s scientific texts are often judged as being of poorer quality than equal contributions authored bymen [99,100]. Authors should substantiate this idea, as the quality of research should be independent of gender or sexual orientation.
Author Response
We would like to thank the Reviewer for their valuable comments as well as their suggestions and detailed instructions on how to improve our manuscript. We are very grateful for all the time and effort expended in reviewing our manuscript. We have attempted to apply and make changes to the manuscript according to your comments. Please find below our detailed responses.
1) Line 89 “Only after accumulating a certain number of scientific publications can an employee be considered for a full professorship [23,24].” Authors should specify, if possible, the minimum number of scientific publications.
Response 1: Thank you for this suggestion. We have added information on the minimum number of scientific publications required for each academic position (Line 86).
2) Line 100 – “Women are less likely than men to hold either the position of full professor 8,37,40,41] or the leadership position [42,43]. Authors must describe the reasons or circumstances for this to happen.
Response 2: In this section (former Line 100) we conclude that the “gender difference in publication activity contributes to the gender differences that are reflected in regard to academic rank” (Line 102). However, we number and discuss other reasons for the gender gap in academic rank in Chapter 4.3. “Future Directions” (Lines 542ff). There, we address how stereotypical ideas about leadership and scientific work being the domain of men can “lead to implicit (often unconscious and unintended) biases that are characterized by women’s scientific contributions being systematically valued less than those of men” (Line 546). Because of those biases women may be disadvantaged in such careers. We focus solely on publication activity in the Introduction, because the current study considered publication activity as an indicator of job performance.
3) Line 481 – “The current study revealed that most employees who experienced high levels of exhaustion took advantage of institutional work” Authors should specify, if possible, in concrete values.
Response 3: Thank you for your suggestion. In order to comply with APA’s Journal Article Reporting Standards (https://apastyle.apa.org/jars/quant-table-1.pdf) we do not report/repeat results in the Discussion section. We refer here to the results reported in Table 4, namely that “work-family services were especially used by employees who experienced frequent exhaustion” (Line 428, Results).
4) Line 507- “Having a certain number of publications with a certain scientific impact is relevant publication because activity is an important criterion for promotion in academic medicine. Authors should specify, in concrete numbers, the number of publications for academic relevance.
Response 4: Based on your Comment 1 we have added this information to the Introduction. In Chapter 2.2.2. of the Methods section we again state the concrete number of publications in order to explain why we formed the chosen categories for the variable publication activity. We do not feel that by repeating this information in the Discussion we add anything further to the argument we want to make. In general, the exact number of publications depends on the respective university and the department. Additionally, when competing for a certain academic position the publication activity and the impact of the competitors is also considered. As a rule of thumb it can be assumed that “the more the better”.
5) Line 546 - “Additionally, women’s scientific texts are often judged as being of poorer quality than equal contributions authored by men [99,100]. Authors should substantiate this idea, as the quality of research should be independent of gender or sexual orientation.
Response 5: We agree with your assertion that research should be independent of gender (and ethnicity, age) or sexual orientation. However, as we explain in this paragraph, “The stereotypical ideas of gender and the perception of a typical scientist being a man often lead to implicit (often unconscious and unintended) biases that are characterized by women’s scientific contributions being systematically valued less than those of men” (Lines 545-550). We list a number of empirical studies that show that the same text, i.e., contribution, was judged differently depending on whether a woman or man was named as the author of the work.
We hope you will find that your comments are reflected in our revised manuscript. We are very grateful for your detailed feedback and instructions on how to improve our manuscript. We have attempted to incorporate all your suggestions. In cases where we were not able to exactly follow your suggestions, we hope you approve of our arguments and approach.
Reviewer 3 Report
This manuscript aims to understand the causal relationships between work stress, exhaustion, and publication activities among medical academic staff. The authors developed hypotheses, conducted a survey among employees at a medical university, and tested the hypotheses using the collected data.
The authors adequately introduced the background related to the research. The descriptions of the data collection and analysis are well-organized and not difficult to follow.
I believe the analysis of the manuscript might be flawed. The problem is that the number of publications and the h-index measure the accumulation of the impacts throughout a researcher's career. However, the work stress and exhaustion are an estimation of the current state. It is possible that an old researcher with extraordinary publication records does not work as hard and does not feel work stress or exhaustion in the recent ten years. However, all his/her impactful publications are associated with work stress and exhaustion when he/her was young. The above example shows that even if we assume a causal relationship between work stress, exhaustion, and publication activity, the analysis described in this paper might not capture it. In addition, whether one person has children younger than 18 is also associated with age. Therefore, I suspect the analysis might be improper, and age might be a confounder that needs to be considered in the analysis. I expect the problems mentioned above might be alleviated if the authors stratify the participants according to their age and redo the analysis.
Author Response
We would like to thank the Reviewer for their valuable comments as well as their suggestions and detailed instructions on how to improve our manuscript. We are very grateful for all the time and effort expended in reviewing our manuscript. We have attempted to apply and make changes to the manuscript according to your comments. Please find below our detailed responses.
Comment: I believe the analysis of the manuscript might be flawed. The problem is that the number of publications and the h-index measure the accumulation of the impacts throughout a researcher's career. However, the work stress and exhaustion are an estimation of the current state. It is possible that an old researcher with extraordinary publication records does not work as hard and does not feel work stress or exhaustion in the recent ten years. However, all his/her impactful publications are associated with work stress and exhaustion when he/her was young. The above example shows that even if we assume a causal relationship between work stress, exhaustion, and publication activity, the analysis described in this paper might not capture it. In addition, whether one person has children younger than 18 is also associated with age. Therefore, I suspect the analysis might be improper, and age might be a confounder that needs to be considered in the analysis. I expect the problems mentioned above might be alleviated if the authors stratify the participants according to their age and redo the analysis.
Response: Thank you very much for raising this concern, for your detailed explanations and for offering concrete suggestions on how to resolve the problem. We agree with you that age is an important variable that needs to be considered. Thus, we have already considered “the control variables of gender, age, employment level, academic position” (Line 383) in our analysis. We report results that include age, employment level and academic position as factors (predictors/control variables) in Table 3 and Table 4. We agree with you that age is associated with the number of publications and the h-index.
We hope you will find that your comments are reflected in our revised manuscript. We are very grateful for your detailed feedback and instructions on how to improve our manuscript. We have attempted to incorporate all your suggestions. In cases where we were not able to exactly follow the suggestions, we hope you approve of our arguments and approach.
Round 2
Reviewer 3 Report
Thank you for the clarifications. My primary concerns were addressed.